# Electrical Detection of Innate Immune Cells

**DOI:** 10.3390/s21175886

**Published:** 2021-09-01

**Authors:** Mahmoud Al Ahmad, Rasha A. Nasser, Lillian J. A. Olule, Bassam R. Ali

**Affiliations:** 1Department of Electrical Engineering, College of Engineering, UAE University, Al Ain 15551, United Arab Emirates; lolule@uaeu.ac.ae; 2Department of Genetics and Genomics, College of Medicine and Health Sciences (CMHS), UAE University, Al Ain 15551, United Arab Emirates; 201670412@uaeu.ac.ae (R.A.N.); bassam.ali@uaeu.ac.ae (B.R.A.)

**Keywords:** dendritic cells, electrical characterization, image processing, immune system, macrophages

## Abstract

Accurately classifying the innate immune players is essential to comprehensively and quantitatively evaluate the interactions between the innate and the adaptive immune systems. In addition, accurate classification enables the development of models to predict behavior and to improve prospects for therapeutic manipulation of inflammatory diseases and cancer. Rapid development in technologies that provide an accurate definition of the type of cell in action, allows the field of innate immunity to the lead in therapy developments. This article presents a novel immunophenotyping technique using electrical characterization to differentiate between the two most important cell types of the innate immune system: dendritic cells (DCs) and macrophages (MACs). The electrical characterization is based on capacitance measurements, which is a reliable marker for cell surface area and hence cell size. We differentiated THP-1 cells into DCs and MACs in vitro and conducted electrical measurements on the three cell types. The results showed average capacitance readings of 0.83 µF, 0.93 µF, and 1.01 µF for THP-1, DCs, and MACs, respectively. This corresponds to increasing cell size since capacitance is directly proportional to area. The results were verified with image processing. Image processing was used for verification because unlike conventional techniques, especially flow cytometry, it avoids cross referencing and by-passes the limitation of a lack of specificity of markers used to detect the different cell types.

## 1. Introduction

Dendritic cells (DCs) and macrophages (MACs) are members of the mononuclear phagocyte system that perform multiple functions during an immune response [1]. Although both DCs and MACs are antigen-presenting cells, they differ in their functions. DCs are specialized in surveillance and the detection of pathogens and, as their name suggests, have elongated structures arising from their body called dendrites [2]. These dendrites increase the surface area of the DCs compared to the cell’s volume [1,3,4]. On the other hand, MACs are mainly involved in the phagocytosis of microbial substances, pathogens, and even cancer cells [5]. MACs also play a significant role in regulating the immune system by releasing cytokines for anti-inflammation [6]. DCs and MACs have been regarded as clearly distinct in terms of cellular function although they occupy overlapping anatomical structures in many body tissues and systems [4]. DCs are stronger in processing antigens and presenting them to the adaptive immune system [7], while MACs are strong in migration to the site at which the pathogen resides and in phagocytosis [8]. DCs and MACs are the key players of the innate immune system as they are the link between the innate and adaptive immune systems [9]. The antigen is captured and processed by these cells and presented to the cells of the adaptive immune system, specifically, the T cells, at specific immunological locations. 

In practice, the process of differentiating between DCs and MACs in vitro is not straightforward [10]. It has heavily relied on cell-surface markers thought to be solely present on one cell type and not on the other [1]. However, growing evidence suggests that many cell surface markers previously used to differentiate between these two cell types overlap [4]. This further complicates our understanding of the mononuclear phagocyte system and confirms the need for a more reliable system to distinguish between these two key immune cell types. Scientists have been using conventional techniques like western blot [11], flow cytometry (FACs) [12], immunohistochemistry [13], and PCR [14] to differentiate between DCs and MACs. Although these techniques are efficient, they are time and money-consuming and also require highly trained technicians. Flow cytometry, the most common technique used in classifying immune cells, depends on detecting cell surface markers present in one cell type and not the other. However, growing evidence suggests that when it is used to compare between DCs and MACs, the markers overlap and display a lack of specificity in comparing the cells, as presented in Figure 1.

Electrical characterization is widely used for the detection and accurate characterization of biological samples [15,16,17]. The last few years have witnessed a substantial growth in new electrical techniques that allow for the detailed study of cells, their characteristics, and functions [15,18,19]. Scientists have focused on studying the cells’ electrical properties due to their relevance in cell activity [17]. These electrical properties are very important because they give insights into the changing biochemical and biophysical properties of the cell that control their interaction with other cells and their interaction with the environment [15].

Over the years, many studies have been conducted to extract biological data from electrical measurements [20]. Useful examples are the resting and membrane potential from the nervous system and the ECG of the heart. Electrical characterization has even expanded to study single cells, viruses, DNA, and even blood samples [18].

Electrical and electrochemical methods have been used widely in several biological applications. Electrical measurements have been used in three different important biological areas: (1) Detection of a disease: measuring the changes in dielectric properties to detect blood in urine samples (hematuria) without the use of inaccurate conventional techniques [21]; (2) characterizing healthy and cancerous cells in different tissue types [22]; and (3) using a label-free tracking method to study the development and progress of living cells in real-time. An example where this was used was to detect the life cycle of budding yeast. The capacitance–voltage dependency was exploited to detect changes in the cell cycle progression [23].

Coupled with electrical characterization is image processing. This has become a vital tool in biological applications for quantifying the phenotypic differences between various cell populations [24]. Screening biological samples has given scientists a deeper insight into the biological systems and their diverse processes such as gene expression, protein modification or interaction, signal transduction, and irregular RNA interference and mutations.

Traditionally, visual analysis is used for image processing. Cells are classified by measurements of cell shape, movement, and protein expression performed manually. This is conducted by suspending cells in a suitable medium, staining them with dye, then analyzing them under a microscope [25]. The manual approach is, however, time-consuming, subjective, and may require a large number of technicians working on the data. Nowadays, image processing is done almost automatically by large processing machines that can deal with high volumes of images, making it faster, more accurate, reliable, and less subjective [26]. Images are visualized as still images, videos, and more recently, 3D and 4D volumetric images. The acquired images can be enhanced by using different fluorescent technologies. The most basic type of analysis is morphological analysis, which does not only refer to metrics of the phenotypical shapes, but also the intensities, the spatial relationships, the staining patterns, and even migration and movement [27].

Automated imaging starts with the principle of extracting the physical parameters of the sample such as the area, density, and morphological properties [28]. Consequently, the data obtained from these images allow the mathematical modeling of biological kinetics and the studying of biochemical signaling networks [29]. The main imaging techniques used for cellular studies are fluorescent microscopy, multiphoton microscopy, atomic and electron microscopy [28]. The fluorescent microscope is mainly used for the visualization of sub-cellular structures and their compartmentalization [30]. It works by capturing the emissions of the excited biological samples using fluorophores. Multiphoton microscopy follows the same principle, but is mainly used for living samples and can image at a deeper scale in comparison to fluorescent microscopy [31]. These techniques have the advantage of high specific identification, but the limitation of photo-bleaching. On the other hand, atomic force microscopy uses Hooke’s law (principle in physics that explains that the force used to compress or extend a spring is proportional to the same distance [32] to acquire the image from the sample [33]). The image is a representation of the forces between the sample and the tip of the probe that scans its surface, and the forces measured vary between chemical, magnetic, electrostatic, and mechanical contact forces. The advantage of this technique is that the sample does not require any special treatment, however, mechanical forces can damage the sample. The last technique, electron microscopy, uses an electron beam to image the object and magnifies it using electromagnetic fields [34]. It provides high resolution but sample preparation takes a long time and cannot be done on living samples.

The data obtained from the image acquisition techniques are processed in software to provide quantitative results [24]. The analysis of the results depends on the advances of the algorithms and processing of the software used. In general, the applications of these software include analyzing the stained tissues, gels, and obtaining the physical and morphological data of the sample [35]. After capturing the sample with the microscope, the software initiates the segmentation process, where the object is located and the boundaries are drawn along the object [36]. The main goal of this process is to simplify the image for quantification. Phenotype quantification is the critical step that follows, the software manages to quantify the image and obtain data such as sample size, distances between the objects, spatial distributions, and in the case of live imaging, tracking the sample movement [2,4]. Phenotypes and data collected from experiments conducted by scientists have also been collected and categorized in shared databases [27]. These databases provide an avenue for users to browse and inquire about experiments and for other scientists to develop more efficient analysis software. Additional experiments like western bot, FACs, and PCR along with the imaging data provide scientists with a better understanding of the biological data.

In this paper, we propose a new, easy, and efficient method to classify immune cells using electrical characterization techniques. The method allows for full differentiation between DCs and MACs. We believe that distinguishing between these cells using electrical characterization supported by image processing will ensure better classification of the innate immune cells during their steady state and inflammatory conditions in different tissues while playing different roles.

## 2. Methods

Two classification approaches are used to distinguish between the different innate immune cells: image processing and electrical characterization. The two approaches are illustrated in Figure 2. Cell differentiation by electrochemical characterization is based on the capacitance values, which are derived from current and voltage readings of cell samples. On the other hand, cell differentiation using image processing is based on analyzing the area, cell count, and morphology of visual data to distinguish the innate immune cells based on their size and morphological differences.

It should be noted that the markers used to specify each type of immune cell are not specific for one type of cell and this leads to the huge drawback of cross-referencing.

Table 1 summarizes the markers used and the specificity for each marker [31,37,38,39,40].

Both experiments began with biological differentiation of cells and their preparation in suspensions. THP-1 was first cultured in RPMI-1640 media, then differentiated into DCs and MACs. Human monocytic THP-1 cell line (ATCC, Manassas, VA, USA) [41] were cultured in RPMI-1640 media supplemented with 10% fetal bovine serum (FBS), 1% sodium pyruvate, 0.01% of mercaptoethanol, and 1% penicillin/streptomycin at 37 °C, 5% CO_2_, and 95% humidity.

Next, cell differentiation was carried out based on the protocol by Berges et al., using the activators specified [42]. For the DCs, THP-1 cells were harvested by centrifugation, then resuspended in culture medium supplemented with 10% FBS at a concentration of 2 × 10^5^ cells/mL and transferred to a final volume of 20 mL into 200-mL tissue culture flasks. To induce differentiation, rhIL-4 (200 ng = 3000 IU/mL), rhGM-CSF (100 ng/mL = 1500 IU/mL), rhTNF-α (20 ng/mL = 2000 IU/mL), and 200 ng/mL ionomycin were added to the FBS- free media. 

For the macrophages, the differentiating and activation protocols of THP-1-derived MACs were adapted and modified from Genin et al. [43]. THP-1 cells were terminally differentiated into uncommitted MACs (MPMA) with 300 nM phorbol 12-myristate 13-acetate (PMA; Sigma-Aldrich, Darmstadt, Germany) in RPMI 1640 media without the FBS supplement. After six hours, differentiating media were removed. The cells were then washed with phosphate-buffered saline (PBS) and rested for 24 h in RPMI 1640 without FBS supplement and PMA. Afterward, cells were activated for 48 h into pro-inflammatory MACs (M_LPS/IFNγ_) by adding 10 pg/mL lipopolysaccharide (LPS; Sigma, St. Louis, MO, USA) and 20 ng/mL IFNγ (Biolegend, San Diego, CA, USA), or into anti-inflammatory MACs (_MIL-4/IL-13_) with 20 ng/mL interleukin 4 (IL-4; Biolegend, USA) and 20 ng/mL interleukin 13 (IL-13; Biolegend, USA).

### 2.1. Flow Cytometry

To validate the differentiation of monocytes, fluorescent surface markers were evaluated using flow cytometry, based on their surface self-antigens. Cultured cells were washed, suspended at 3 × $10^{4}$ in 200 µL cold FACS solution (DPBS; Gibco-Invitrogen, San Diego, CA, USA) and incubated with FITC- or PE-conjugated monoclonal antibodies or appropriate isotypic controls for 30 min. Cells were then washed twice and resuspended in 300 µL of cold FACS solution. Stained cells were analyzed with (BD Accuri C6 plus). Cell debris was excluded from the analysis by setting a gate on forward and side scatter that included only cells that were viable. Results were processed using FlowJo Software (version 7).

### 2.2. Image Acquisition and Processing

The image processing method consists of analyzing the cells based on visual data supported by their morphological and structural differences. Images were captured using an Olympus Fluorescent Microscope and quantified using ImageJ software (National Institute of Health, Gaithersburg, MD, USA) [44]. The software was used to obtain the ratio of THP-1 to DCs, THP-1 to MACs, and the average area of the three types of cells. ImageJ software segments the images, recognizes the cells, differentiates between the different types of cells, and automatically calculates the area.

### 2.3. Electrochemical Measurement

For the electrochemical approach, measurements were performed using the µSTAT 400 potentiostat (Metrohm DropSens, Oviedo, Spain) [45]. This was a portable BiPotentiostat/Galvanostat with maximum measurable current and potential of ±40 mA and ±4 V, respectively. It can be used for voltammetric, amperometric, or potentiometric measurements. It has connectors that allow for connection to screen printed or coaxial electrodes and can be used with a one- or two-working electrode configuration. It connects to a PC via USB or Bluetooth.

All measurements were carried out at room temperature. The electrochemical measurements were controlled using Dropview software. Prior to the experiments, two optimizations were performed: (1) identify the optimum step potential (Estep) and scan rate (S_rate_); and (2) determine the best electrode option between the chip and coaxial cable.

#### 2.3.1. System Optimization for E_step_ and S_rate_

To find the optimum of S_rate_, the voltage was swept from −0.9 V to 0.9 V while E_step_ was kept constant at 0.002 V and the S_rate_ was varied from 0.004 V/s to 2 V/s. An optimum S_rate_ of 0.004 V/s was selected, which allowed for accurate data (this value of S_rate_ limits the non-Faradic current and therefore background noise, which affects the sensitivity of the voltammetry system [46]), sufficient current flow, and absence of time-dependent charging and discharging effects. This value gave the highest capacitance resolution, which can aid with distinguishing between cells.

Second, both E_step_ and S_rate_ values were varied simultaneously from 0.009 V to 0.01 V and from 0.009 V/s to 2 V/s, respectively. It was found that corresponding low values did not allow for proper current flow and high values of S_rate_ did not allow for sufficient charge of the sample. Additionally, equal values of E_step_ and S_rate_ did not provide the correct shape for the cyclic voltammogram. Hence, from the experiments, the optimum values of E_step_ and S_rate_ were selected as 0.002 V and 0.04 V/s, respectively.

#### 2.3.2. System Optimization for Electrode Selection

The screen printed electrode was tested for its performance. It comprised three electrodes: a working electrode, reference electrode and counter electrode. The sample was applied to all electrodes and then the electrode was connected to the DropSens machine via a port with silver contacts. It was found that although the screen printed electrode is low cost, disposable, and can give results for low volumes, current flow in the samples experienced interference, and as a result, not all cells were charged. Instead, a coaxial cable was used. The coaxial cable is easy to clean between trials and most importantly, guarantees equal current flow throughout the sample.

Using the coaxial cable, the DropSens machine was configured for two electrode measurements with one electrode used as the working electrode and the other electrode used as the reference/counter electrode. The cable is an open ended coaxial adaptor with inner and outer conductor electrode dimensions of 2 mm and 5 mm, respectively, and a length of 7 mm, which allows for a sample volume of 500 µL. Both electrodes are made from Nickel. The coaxial cable was secured to ensure stability during measurements. The electrolyte used was the RPMI full media supplemented with 10% FBS.

#### 2.3.3. Measurement Procedure

Once optimization was completed, cyclic voltammetry measurements were performed between 0.9 V to −0.9 V, E_step_ of 0.002 V and S_rate_ of 0.04 s per step using the coaxial cable. Cells were prepared using RPMI full media supplemented with 10% FBS. After the activation process, cells were centrifuged and prepared at different dilutions from 10 to 10^5^ per 500 µL. This was carried out by first counting the cells using a hemocytometer, then diluting them to the necessary concentrations. Data were extracted directly from drop view using the cyclic voltammetry technique. The results exported were current vs. voltage.

After extracting the current vs. voltage data, the capacitance of the biological cells was determined using MATLAB code based on the fact that the capacitive current is proportional to the rate of change of the potential with the constant of proportionality equal to the capacitance, as shown in Equation (1).
(1)$$i\left(t\right)=C\frac{dv\left(t\right)}{dt}$$
where Q(t) is the time-dependent charge; C is the capacitance in farads; and V(t) is the time dependent voltage in volts.

### 2.4. Statistical Analysis

All measurements were performed at least three times, and the results represent the mean ± standard deviation. A two-tailed Student’s *t*-test with a significance level of 0.05 was also performed.

## 3. Results and Discussion

The main goal of this work was to find a way to identify immune cells without the drawback of cross-referencing. For flow cytometry, cells were selected by a gating process. Debris were excluded and only stained cells were selected. The results are plotted in the histogram shown in Figure 3. Cell surface markers for CD83, CD197, HLA-DR, CD1c, and CD11c expression on THP-1 cells and the differentiated DCs and MACs were analyzed. Two sample t tests were performed with a *p*-value of 0.05. The *p*-values are tabulated in the Appendix A in Table A1.

To begin with, CD83 represents an important marker that is specific for DCs. However, our results showed that there is no significant difference between DCs and THP-1 or MACs, and this is supported by a study undertaken by D. Ferenbach and J. Hughes and others [4,47]. On the other hand, CD197 expression only showed differences between MACs against THP-1 and DCs against THP-1. This can be attributed to CD197 being a marker for antigen presenting cells, however, it cannot classify between the different types of antigen presenting cells. Regarding HLA-DR marker expression, it presented on all the three types of immune cells [37,38], hence, we could see no difference with the flow cytometry results. CD1c is a marker for DCs, this is supported by our results as they can classify DCs from MACs, but not from THP-1 cells. However, CD11c is a marker for all three cells [39] and as per our results, there were no differences between these cells, using this marker. Hence flow cytometry analyzes the data by giving statistical significance to values but fails to interpret it into biological significance, thus failing to give an identity to the immune cells [31]. 

The morphology and structure of the three types of immune cells identified using the image segmentation approach for electrical characterization are demonstrated in Figure 4. The THP-1 cells can be easily distinguished from DC by their round structure without elongations. Once activated, the non-adherent THP-1 cells differentiate to adherent cells that are morphologically different from their inactive forms. On the other hand, MACs and DCs take more space to spread out due to the larger size of the former and the presence of dendrites in the latter, as shown in Figure 4.

Figure 4D–F shows the detailed selection of immune cells using the software. The software highlights the morphological differences (it marks the outside border of the cell yellow). After the selection of each cell, the software automatically calculates the area of the cell. Results were obtained from three different images to statistically compare the area of each cell. Figure 5 shows a summary of the results. The averages were obtained for measurements conducted on 200 cells of each type. The MACs were found to have the largest area and the THP-1s were the smallest due to their rounded shape. These results are supported by findings in the literature [48,49].

For electrochemical characterization, the DropSens technology was used to obtain the I–V curves for the three immune cells. The results are shown in Figure 6 for different cell concentrations. The current versus time and voltage versus time results are shown in Figure 7. When the positive voltage is applied, the cell suspensions begin to oxidize near the working electrode, this results in an increase in anodic current. This occurs until a peak potential of 0.9 V, wherein a peak anodic current is recorded. After this, a reductive scan is applied, that is, the applied potential is reduced, causing a re-reduction of the oxidized suspension. In other words, the reducing potential now results in a cathodic current (increasingly negative current). At a maximum negative potential of −0.9 V, the maximum cathodic current is recorded (maximum negative current). Although reduction peaks at −0.2 V were observed for all experiments, the regions of maximum and minimum potential were of more interest because the peaks corresponded to the sample concentrations [46]. The peak anodic and cathodic currents had equal magnitude and opposite sign. As the potential is increased positively again, the oxidation and increasing flow of anodic current repeats.

However, since the voltammogram results showed no significant difference within the three types of cells hence, the capacitance was pursued as a means of identification and differentiation. Capacitance measurements have been shown to be a reliable marker for tracking cell surface area and therefore cell size [50]. The graphs of the extracted capacitance are shown in Figure 8.

Comparing the three plots, it was noticed that only THP-1 cells displayed the expected trend of increased capacitance with increasing concentration. Electrochemical sensors react with the analyte under test to produce an electrical signal proportional to the analyte concentration [51]. The inconsistency with DCs and MACs was likely due to the lack of a homogenous suspension as cells might not have fully differentiated. Therefore, to obtain a better picture of the capacitance data, the value of the media was de-embedded from the other samples, that is, each concentration value was divided by the corresponding media value. Additionally, because electronic measurements of conductive solutions are often affected by ionic effects like electrode polarization that occurs within the Debye screening length of the solution, de-embedding can mitigate this effect since the electrode polarization is localized and remains constant for a particular ion concentration and device geometry [52]. Figure 9 displays the data for the three immune cells after the de-embedding process.

From the initial capacitance plot (Figure 8), it was seen that the capacitance peaked at about 29.2 s for all experiments. Therefore, the values of the capacitance for this time measurement were extracted from the de-embedded data and compared as shown in Figure 10. As expected for each cell type, there was a general increase in capacitance with concentration. This is illustrated in Figure 10A. This is attributed to the fact that an increase in the number of cells results in an increase in total surface area and since the area is directly proportional to capacitance, an increase in capacitance is observed. Although a clear distinction between the MACs and DCs can be seen (the MACs have a larger capacitance and therefore are larger and the DCs have a lower capacitance and therefore are smaller) to more clearly differentiate between all three cell types and by-pass the inconsistency at the 10^5^ concentration, the average capacitance for three concentrations was plotted as shown in Figure 10B. It should be noted that the reason for the discrepancy at 10^5^ was attributed to errors in pipetting or sample preparation. It is therefore recommended that several concentrations be used for proper validation. Additionally, more accurate results can be obtained by using polished, well cleaned electrodes and smaller sample volumes for greater sensitivity.

The results showed that the lowest average value of capacitance was for THP-1 (0.83 µF), followed by DCs (0.93 µF), and finally, the largest capacitance was reported for the MACs (1.01 µF). This corresponds to an increasing cell size from THP-1 to DCs to MACs, consistent with the results reported in Figure 6 and in the literature. Although from the results the distinction is possible with only the lowest concentration, the authors recommend the use of the three lowest concentrations used in this paper at a minimum. These concentration ranges are comparable to those used for flow cytometry, for example, Bio-Rad recommends concentrations of 10^5^–10^7^ cells/mL [53].

The assay described in this study can be practically functionalized by creating a compact battery powered and/or directly powered sensing unit and a control unit. The sensing unit will comprise two electrodes separated by a gap into which the specimen can be loaded via pipette. When voltage is applied to the electrode, the corresponding resultant current can be measured by the electrodes. The sensing unit will connect to the control unit where voltage value and step size can be controlled or swept. Once cyclic voltammetry measurements are performed, software in the control unit can perform further processing on the extracted current and voltage data to calculate the capacitance of the sample under test. The results could then be displayed in the control unit graphical user interface or to a PC via USB/wirelessly for further processing.

## Figures and Tables

**Figure 1 sensors-21-05886-f001:**
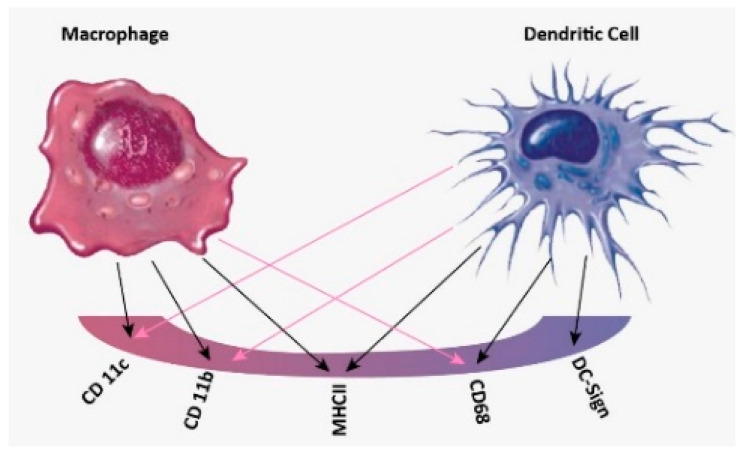
Overlapping of cell surface markers between MACs and DCs. DCs and MACs share the same surface markers CD11c, CD11b, MHCII, CD68.

**Figure 2 sensors-21-05886-f002:**
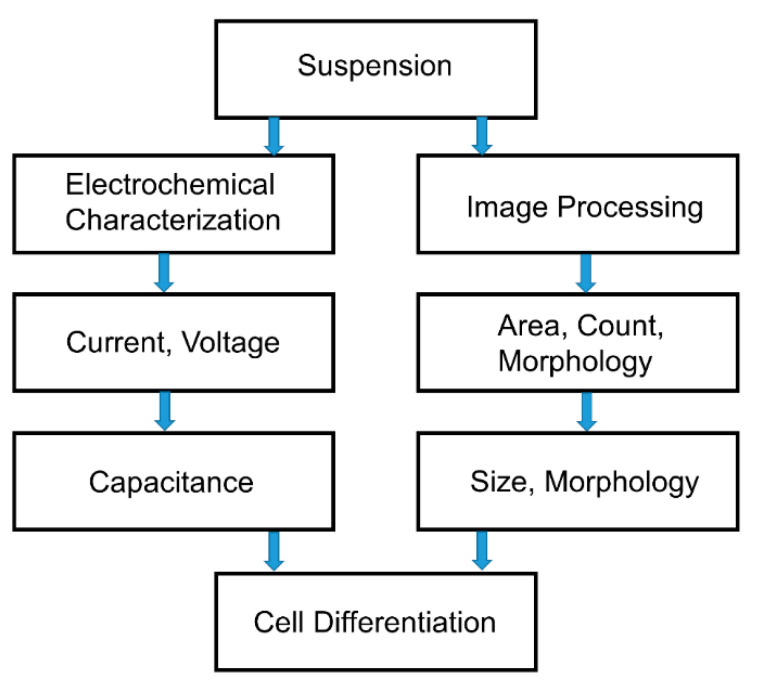
The two approaches for innate immune cell differentiation.

**Figure 3 sensors-21-05886-f003:**
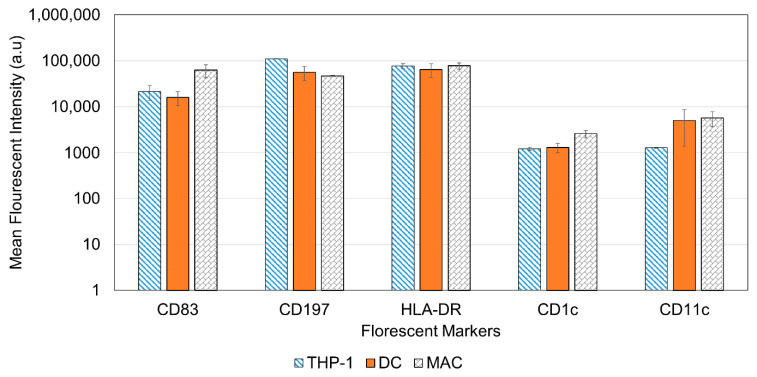
Average mean fluorescent intensity of different cell markers for THP-1, DCs, and MACs with S.E.M bars obtained for three measurements. Cultured cells were washed, suspended at 3 × 10^4^ in 200 µL cold FACS solution (DPBS; Gibco-Invitrogen) and incubated with FITC- or PE-conjugated monoclonal antibodies or appropriate isotypic controls for 30 min. Cells were then washed twice and resuspended in 300 µL of cold FACS solution. Stained cells were analyzed with BD Accuri C6 plus. Cell debris was excluded from the analysis by setting a gate on forward and side scatter that included only cells that are viable.

**Figure 4 sensors-21-05886-f004:**
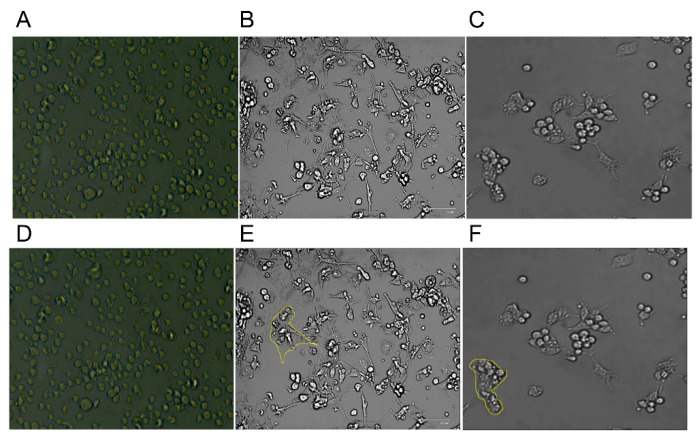
(**A**) THP-1 Immune cells before differentiation. THP-1 was first cultured in RPMI-1640 media, then differentiated into DCs and MACs. Human monocytic THP-1 cell line (ATCC, Manassas, VA, USA)35 were cultured in RPMI-1640 media supplemented with 10% fetal bovine serum (FBS), 1% sodium pyruvate, 0.01% of mercaptoethanol, and 1% penicillin/streptomycin at 37 °C, 5% CO_2_, and 95% humidity. (**B**) DCs and (**C**) MACs after differentiation, respectively. DCs were differentiated based on the Berges et al. protocol. To induce differentiation rhIL-4 (200 ng = 3000 IU/mL) and rhGM-CSF (100 ng/mL = 1500 IU/mL), rhTNF-α (20 ng/mL = 2000 IU/mL), and 200 ng/mL ionomycin were added to the FBS-free media. For the macrophages, the differentiating and activation protocols of THP-1-derived macrophages were adapted and modified from Genin et al. [37]. THP-1 cells were terminally differentiated into uncommitted macrophages (MPMA) with 300 nM phor-bol 12-myristate 13-acetate (PMA; Sigma-Aldrich, Germany) in RPMI 1640 media without FBS supplement. Afterward, cells were activated for 48 h into pro-inflammatory macrophages (MLPS/IFNγ) by adding 10 pg/mL lipopolysaccharide (LPS; Sigma, USA) and 20 ng/mL IFNγ (Biolegend, San Diego, CA, USA), or into anti-inflammatory macrophages (MIL-4/IL-13) with 20 ng/mL interleukin 4 (IL-4; Bio-legend, USA) and 20 ng/mL interleukin 13 (IL-13; Biolegend, USA). THP-1 cells have a round shape and are suspended in the media, DCs are attached and spread their dendrites in the flask. MACs are also adherent, but without the elongations of the DCs. (**D**–**F**) show the selection undertaken in ImageJ software for the calculation of the area of the THP-1, DCs, and MACs, respectively.

**Figure 5 sensors-21-05886-f005:**
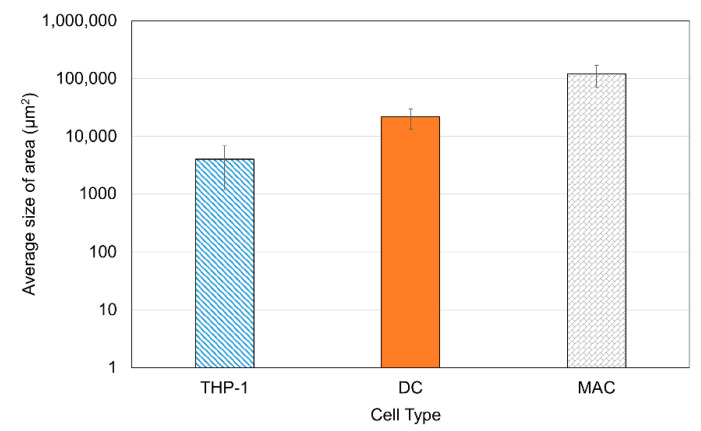
The calculated average area of each cell with S.E.M bars. MACs have the largest area, followed by DCs, and finally THP1 cells.

**Figure 6 sensors-21-05886-f006:**
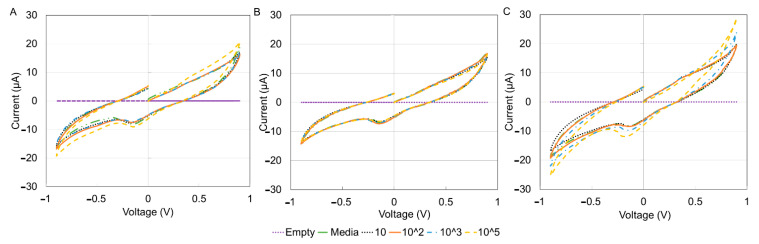
I–V curve for the three types of cells using drop sense technology. (**A**) THP1, (**B**) DCs, (**C**) MACs. There were no clear differences between the three graphs. RPMI full media supplemented with 10% FBS was used to dilute the cells. It was also used as the media. Measurements were conducted using a two nickel electrode configuration, scan range of −0.9 V to 0.9 V and a scan rate of 0.04 V/s.

**Figure 7 sensors-21-05886-f007:**
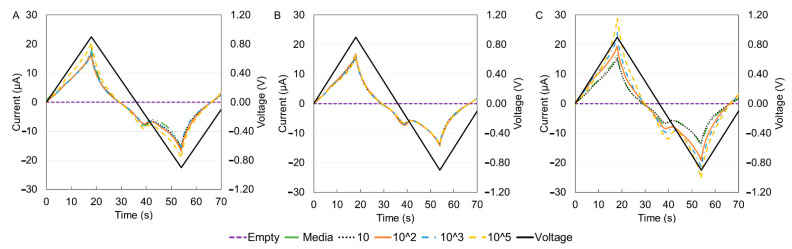
Current versus time and voltage versus time curves for the three types of cells from the drop sense technology. (**A**) THP-1, (**B**) DCs, (**C**) MACs. RPMI full media supplemented with 10% FBS used to dilute the cells. It was also used as the media. Measurements were conducted using a two nickel electrode configuration, scan range of −0.9 V to 0.9 V and a scan rate of 0.04 V/s.

**Figure 8 sensors-21-05886-f008:**
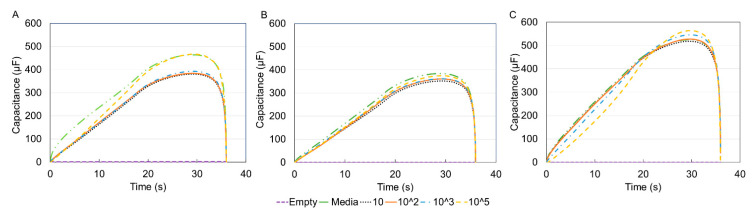
Capacitance–time curve for the three types of cells before media de-embedding (removing the value of media from the rest of the samples) (**A**) THP-1, (**B**) DCs, (**C**) MACs. Capacitance values were extracted using MATLAB, based on the fact that the capacitive current measured is proportional to the rate of change of the applied potential with the constant of proportionality equal to the capacitance. There is no consistent trend between the concentration and capacitance.

**Figure 9 sensors-21-05886-f009:**
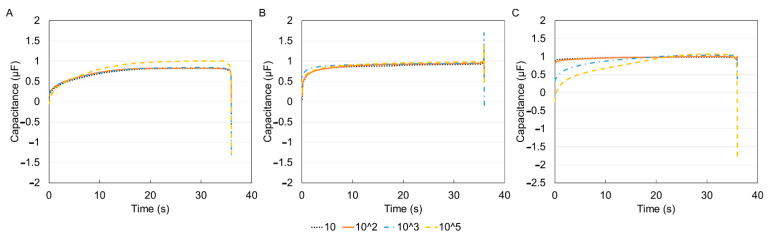
Capacitance–time curve for the three types of cells after media de-embedding (**A**) THP-1, (**B**) DCs, (**C**) MACs. De-embedding was performed by diving each of the concentration values in Figure 8 by their corresponding media value.

**Figure 10 sensors-21-05886-f010:**
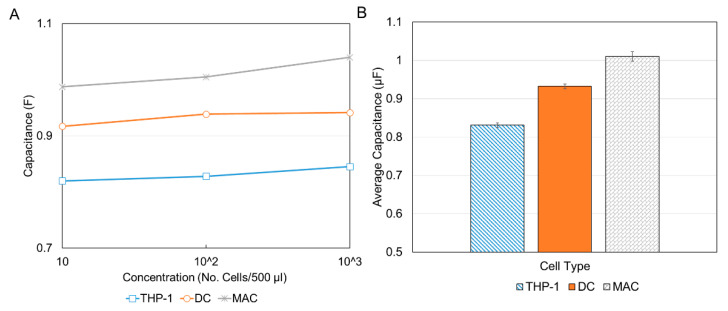
Capacitance vs. concentration for the three types of cells after the de-embedding process. (**A**) Capacitance versus concentration at 29.2 s where maximum capacitance occurs for each of the cell types. (**B**) Average capacitance for the three concentrations with S.E.M error bars. MACs had the highest values and DCs had the lowest values, consistent with the literature.

**Table 1 sensors-21-05886-t001:** Markers used for immune cells and their specifications.

Marker	Specificity	Ref.
CD83	Marker for mature DCs and very weak for THP-1	[31]
CD197	Receptor for T-cells, B cells, Natural killer cells and DCs	[37]
HLA-DR	Recognizes T cells, DCs, MACs, and B cells	[38]
CD1c	Subset of B cells and DCs	[39]
CD11c	For monocytes, MACs, DCs, Natural killer cells, T and B cells	[40]

## Data Availability

Data is contained within the article.

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
