# Peer review of "Electrical Detection of Innate Immune Cells"

_sensors, 2021, doi:10.3390/s21175886_

Round 1
Reviewer 1 Report
This work describes a new immunophenotyping technique that uses electrical features to distinguish dendritic cells (DCs) from macrophages. The electrical characteristics are based on capacitance measurements, which are a reliable marker of cell surface area and therefore of cell size. We differentiated THP-1 cells into DCs and macrophages in vitro and performed electrical measurements on these three cell types. The results showed that the average capacitance readings for THP-1, DCs and macrophages were 0.87 μF, 0.94 μF and 1.03 μF, respectively. this corresponds to an increase in cell size, as capacitance is proportional to area. The results were validated using image processing. This assay is creative, but it can hardly be considered as belonging to the field of Biosensor, so I think the authors should choose another journal.
- First, the Experimental section should be placed after Introduction.
- All electrochemical measurements were controlled using Dropview software without any specific sensor design. The authors have only optimized the parameters of the test.
- Based on the error bars measured in Figure 10, it is difficult to demonstrate that the proposed method can identify the three cell typologies with great accuracy.
- How many of the cells were counted by the authors using ImageJ to arrive at the average area?
Author Response
Reviewer #1
This work describes a new immunophenotyping technique that uses electrical features to distinguish dendritic cells (DCs) from macrophages. The electrical characteristics are based on capacitance measurements, which are a reliable marker of cell surface area and therefore of cell size. We differentiated THP-1 cells into DCs and macrophages in vitro and performed electrical measurements on these three cell types. The results showed that the average capacitance readings for THP-1, DCs and macrophages were 0.87 μF, 0.94 μF and 1.03 μF, respectively. this corresponds to an increase in cell size, as capacitance is proportional to area. The results were validated using image processing. This assay is creative, but it can hardly be considered as belonging to the field of Biosensor, so I think the authors should choose another journal.
Response: We appreciate Reviewer#1 for his comments and concerns. We would like to thank him for considering our technique as a new and creative. Indeed the assay is a novel immunophenotyping technique. Furthermore the paper definitely does present a bio-sensing mechanism that has been tested using an experimental setup. The sensing concept is presented and the relevant sensor can be easily built using two electrodes separated by a gap to load the specimen. To elaborate on this we have added the following:
“The assay described in this study can be practically functionalized by creating a battery powered and/or directly powered sensing unit and a control unit. The sensing unit will comprise two electrodes separated by a gap into which the specimen can be loaded via pipette. When voltage is applied to the electrode the corresponding resultant current can be measured by the electrodes. The sensing unit will connect to the control unit where voltage value and step size can be controlled or swept. Once cyclic voltammetry measurements are performed, software in the control unit can perform further processing on the extracted current and voltage data to calculate the capacitance of the sample under test. The results could then be displayed in the control unit graphical user interface or transferred to a PC via USB/wirelessly for further processing.” (Page 13, line 405-413)
- First, the Experimental section should be placed after Introduction.
Response: Experimental section (Methods) has been placed after introduction (Page 4).
- All electrochemical measurements were controlled using Dropview software without any specific sensor design. The authors have only optimized the parameters of the test.
Response: Although no specific sensor was designed, the paper shows an application of using the Dropsen platform as a sensing device as well as image processing for characterization which we believe falls under the sensing systems, and sensing and imaging scope highlighted in the journals aims and scope. Additionally a description of the practical sensor functionalization has been added (Page 13, line 405-413)
- Based on the error bars measured in Figure 10, it is difficult to demonstrate that the proposed method can identify the three cell typologies with great accuracy.
Response: The error bars in the original figure were set to the default 5%. This has been corrected to standard error bars which take into account the standard deviation. The corrected error bar length and non-overlap are indicative of the accuracy of our proposed method. (Page 12, Figure 10). Recommendations were also made for more accurate results: “Additionally more accurate results can be obtained by using polished well cleaned electrodes and smaller sample volumes for greater sensitivity.” (Page 12, line 388-380)
- How many of the cells were counted by the authors using ImageJ to arrive at the average area?
Response: The cell count for the different cells using ImageJ has been included in the manuscript. “The averages were obtained for measurements done on 200 cells of each type.” (Page 9, line 317-318).
Reviewer 2 Report
In this work, Ahmad et al., report a novel immune-phenotyping technique using electrical characterization to differentiate between the two most important cell types of the innate immune system such as dendritic cells (DCs) and macrophages. It is an interesting work where authors provide a new strategy for the electrical detection of innate immune cells. In my opinion, the manuscript can be published in this journal, after some critical points should be clarified. The following issues should be addressed.
Major issues
- First, from the results, cell confluence seems to have affected the capacitance result, but it seems that the experiment was not conducted by controlling the cell confluence between different cells. These experiments are very important experiments to determine whether this technology is available. Therefore, it is difficult to prove the effectiveness of this technique unless comparative experiments are performed after cell confluence is equally controlled in all cell types.
- For electrochemical characterization, the author used new DropSens technology to obtain I-V curves. What is the new DropSens technology? Authors should clearly describe this technique.
- In figure 10b, the authors reported that the average capacitance of DC was higher than that of THP-1. However, the capacitance of THP-1 was higher than that of DC at 105 concentrations (Figure 10b). Authors should explain the reasons for the conflicting results.
- The concentration range used to identify differences between cells was 10-105. It should be stated how many concentrations could be used to identify the difference in the capacitance of cells by the technique presented by the authors. Additionally, the author should compare the capable cell concentration range between this technology between conventional equipment such as flow cytometry.
Minor issues
- Some references in the introduction are missing. Cite appreciate references in the following sentences in the introduction.
Author Response
Reviewer #2
In this work, Ahmad et al., report a novel immune-phenotyping technique using electrical characterization to differentiate between the two most important cell types of the innate immune system such as dendritic cells (DCs) and macrophages. It is an interesting work where authors provide a new strategy for the electrical detection of innate immune cells. In my opinion, the manuscript can be published in this journal, after some critical points should be clarified. The following issues should be addressed.
Response: We thank respected Reviewer#2 for his comments and concerns. We would like to thank him for recognizing our technique novel and interesting.
Major issues
- First, from the results, cell confluence seems to have affected the capacitance result, but it seems that the experiment was not conducted by controlling the cell confluence between different cells. These experiments are very important experiments to determine whether this technology is available. Therefore, it is difficult to prove the effectiveness of this technique unless comparative experiments are performed after cell confluence is equally controlled in all cell types.
Response: For the capacitance experiments, and equal number of each cell type were counted using a Hemocytometer. The concentrations were then controlled by the various levels of dilution from 10 to 105 per 500 µl. This has been clarified in the manuscript. (Page 7, line 242 -243).
- For electrochemical characterization, the author used new DropSens technology to obtain I-V curves. What is the new DropSens technology? Authors should clearly describe this technique.
Response: Statement has been revised to “For electrochemical characterization, the DropSens technology was used…” The word new has been removed (Page 10, line 324). All measurement procedures have been discussed in the methods section. A brief description of the DropSens machine has been included under the methods section. “It is a portable BiPotentiostat/Galvanostat with maximum measurable current and potential of ±40 mA and ±4 V respectively. It can be used for Voltammetric, Am-perometric or Potentiometric measurements. It has connectors that allow for connec-tion to screen printed or coaxial electrodes and can be used with one- or two-working electrodes configuration. It connects to a PC via USB or Bluetooth.” (Page 6, line 198-202).
- In figure 10b, the authors reported that the average capacitance of DC was higher than that of THP-1. However, the capacitance of THP-1 was higher than that of DC at 105 concentrations (Figure 10b). Authors should explain the reasons for the conflicting results.
Response: Explanation included in the manuscript; “It should be noted that the reason the discrepancy at 105, is attributed to errors in pi-petting or sample preparation. It is therefore recommended that several concentrations be used for proper validation. Additionally more accurate results can be obtained by using polished well cleaned electrodes and smaller sample volumes for greater sensitivity. ” (Page 11, line 386-389)
- The concentration range used to identify differences between cells was 10-105. It should be stated how many concentrations could be used to identify the difference in the capacitance of cells by the technique presented by the authors. Additionally, the author should compare the capable cell concentration range between this technology between conventional equipment such as flow cytometry.
Response: Recommendations included in manuscript; “Although from the results, the distinction is possible with only the lowest concentration, the authors recommend the use of the three lowest concentrations used in this paper at a minimum. These concentration ranges are comparable to those used for flow cytometry for example Bio-Rad recommends concentrations of 105-107 cells/ml40.” (Page 11, lines 394-395)
Minor issues
- Some references in the introduction are missing. Cite appreciate references in the following sentences in the introduction.
Response: Please advise on the sentences that require referencing.
Reviewer 3 Report
The authors have reported on Electrical Detection of Innate Immune Cells. The approach is interesting. However, there are a few concerning issues to be addressed:
- Introduction:
- There is no need to provide details shown in Table 1 and Figure 1. They are not directly relevant for this work. They can be mentioned very briefly in a sentence or two within the text, but not much more.
- Inadequate references. For example, after (i) Line 52 ‘It has heavily …’; (ii) Line 67; (iii) Line 69;
- Include more information on image processing… From visual to current technology.
- Results and Discussion
- Table 2 should be under methods
- Table 3 can not be stand alone. It should be removed. It should be either included as supporting information, or the statistics could be described briefly within the manuscript.
- Figure 4 legend: It could be made clearer by adding ‘respectively’ or rephrase entirely. It is not immediately clear which image is what.
- Line 149: ‘Results were obtained from different images’. How many images? Please include the information.
- Line 157: ‘When….cell start to get oxidised…’. The authors should attempt to explain what is being oxidised. Not all cell components get oxidised at some specific/prescribed conditions. What are the conditions? What type of electrode system is being used? What about the electrolyte? pH conditions, etc. All these affect redox activity of molecules. These conditions would be discussed well in the methods section. The authors only mentioned SPE vs coaxial cable. What was the WE, RE, CE? What about surface area of the electrode systems? ‘Electrical’ detection is the main part of this paper. These important parameters and thorough discussion of the results should follow.
- Figures 4 and 5 could be combined into 1. It’s about surface area coverage of the cells. Are these individual (single) cells or a cluster of cells? From the images in Figure 4, it looks like a cluster of cells (I have no experience here). If it’s a cluster, how is the size of the cluster determined? For example, a cluster of 5 cf with a cluster of 10. How would this affect the results and their interpretation?
- Lines 161 and 164: I don’t understand the way the data is being discussed. Why is 0.9V described as maximum positive (and negative) potential? What I see is a reduction peak potential at approximately -0.2V
- Line 183: ‘…display a trend’. I don’t follow this either. All the results show a similar pattern. Perhaps the authors would elaborate and rephrase to avoid confusion.
- Two many figures. Perhaps some of this could be sent to supporting information.
- Line 277: Please clarify ‘accurate data’.
- Number of replicates and what error bars stand for (SEM or CV or Std) should be included in Figure captions.
- Figure legends should be stand alone. Please include experimental parameters and conditions in captions .
- Consistency is terminology. For example, use DC consistently, instead of dendritic or dendritic cells.
- Methods
- Electrochemical methods have not be well discussed, especially the electrochemical system itself (electrode types, surface areas, electrolyte and similar).
- Please present system optimisation data, at least in SI.
- Line 276: The presentation of the results is suboptimal. The results are left hanging, no proper units used. For example, scan rate data should be presented as 0.004V/s. (or 4mV/s). Importance of optimising step potential as well as scan rate should be mentioned.
- Line 284: Change ‘voltammeter graph’ to ‘voltammogram’.
- Line 292: ‘… being charged’. Please rephrase. Technically reads poorly.
Author Response
Reviewer #3
The authors have reported on Electrical Detection of Innate Immune Cells. The approach is interesting. However, there are a few concerning issues to be addressed:
- Introduction:
- There is no need to provide details shown in Table 1 and Figure 1. They are not directly relevant for this work. They can be mentioned very briefly in a sentence or two within the text, but not much more.
Response: We thank respected Reviewer#3 for his comments and concerns. We would like to thank him for acknowledging the interesting approach of our technique. The details given in Table 1 and Figure 1 were already mentioned in the text. The table and figure were added to more clearly highlight the differences in cell functions and therefore justify the need for cell differentiation. However as per the reviewer’s suggestion they have been removed.
- Inadequate references. For example, after (i) Line 52 ‘It has heavily …’; (ii) Line 67; (iii) Line 69;
Response: The associated references have been added to the necessary lines.
“It has heavily relied on cell-surface markers thought to be solely present on one cell type and not on the other1….Electrical characterization is widely used for the detection and accurate characterization of biological samples15–17. The last few years have witnessed a substantial growth in new electrical techniques that allow for the detailed study of cells, their characteristics and functions15,18,19.” (Page 2, line 45; Page 3, line 61-64)
- Include more information on image processing… From visual to current technology.
Response: Additional discussions on image processing has been included in the introduction, highlighting the development from visual to the current technology.
“Traditionally, visual analysis is used for image processing. Cells are classified by measurements of cell shape, movement, protein expression performed manually. This is done by suspending cells in a suitable medium, staining them with dye then analyzing them under a microscope25. The manual approach is, however, time-consuming, subjective and may require a large number of technicians working on the data. Nowadays image processing is done almost automatically by large processing machines that can deal with high volumes of images, making it faster, more accurate, reliable and less subjective26. Images are visualized as still images, videos, and more recently 3D and 4D volumetric images. The acquired images can be enhanced by using different fluorescent technologies. The most basic type of analysis is the morphological analysis that does not only refer to metrics of the phenotypical shapes but also the intensities, the spatial relationships, the staining patterns and even migration and movement27.
Automated imaging starts with the principle of extracting the physical parameters of the sample such as the area, density and morphological properties28. Consequently the data obtained from these images allow the mathematical modeling of bio-logical kinetics and the studying of biochemical signaling networks29. The main imaging techniques used for cellular studies are fluorescent microscopy, multiphoton microscopy, atomic and electron microscopy28. The fluorescent microscope is mainly used for the visualization of the sub-cellular structures and their compartmentalization30. It works by capturing the emissions of the excited biological samples using fluorophores. Multiphoton microscopy follows the same principle but is mainly used for living samples and can image at a deeper scale in comparison to the fluorescent microscopy31. These techniques have the advantage of high specific identification but the limitation of photo-bleaching. On the other hand atomic force microscopy uses Hooke’s law (principle in physics that explains that the force used to compress or extend a spring is proportional to the same distance32 to acquire the image from the sample33. The image is a representation of the forces between the sample and the tip of the probe that scans its surface, the forces measured vary between chemical, magnetic, electrostatic and mechanical contact forces. The advantage of this technique is that the sample does not require any special treatment, however, mechanical forces can damage the sample. The last technique, the electron microscopy uses an electron beam to image the object and magnifies it using electromagnetic fields34. It provides high resolution but sample preparation takes a long time and it cannot be done on living samples.
The data obtained from the image acquisition techniques is processed in software to provide quantitative results24. The analysis of the results depends on the advances of the algorithms and processing of the software used. In general the applications of these software include analyzing the stained tissues, gels and getting the physical and morphological data of the sample35. After capturing the sample with the microscope, the software initiates the segmentation process, where the object is located and the boundaries are drawn along the object36. The main goal of this process is to simplify the image for quantification. Phenotypes quantification is the critical step that follows, the soft-ware manages to quantify the image and get data like, sample size, distances between the objects, spatial distributions and incase of live imaging tracking the sample move-ment24. Phenotypes and data collected from experiments conducted by scientists have also been collected and categorized in shared databases27. These databases provide an avenue for users to browse and inquire about experiments and for other scientists to develop the more efficient analysis software. Additional experiments like western bot, FACs, PCR along with the imaging data give scientists a better understanding of the biological data.” (Page 3, lines 86-132)
- Results and Discussion
- Table 2 should be under methods
Response: The table has been moved to the methods section as suggested. (Page 5, Table 1)
- Table 3 cannot be stand alone. It should be removed. It should be either included as supporting information, or the statistics could be described briefly within the manuscript.
Response: The table has been moved to Supplementary material (Page 13, lines 421- 428, Table A1)
- Figure 4 legend: It could be made clearer by adding ‘respectively’ or rephrase entirely. It is not immediately clear which image is what.
Response: Figure 4 caption has been revised. See point 15. (Page 9)
- Line 149: ‘Results were obtained from different images’. How many images? Please include the information.
Response: This information was included in the manuscript. “Results were obtained from three different images to statistically compare the area of each cell.” (Page 9, lines 316-317)
- Line 157: ‘When….cell start to get oxidised…’. The authors should attempt to explain what is being oxidised. Not all cell components get oxidised at some specific/prescribed conditions. What are the conditions? What type of electrode system is being used? What about the electrolyte? pH conditions, etc. All these affect redox activity of molecules. These conditions would be discussed well in the methods section. The authors only mentioned SPE vs coaxial cable. What was the WE, RE, CE? What about surface area of the electrode systems? ‘Electrical’ detection is the main part of this paper. These important parameters and thorough discussion of the results should follow.
Response: Experimental conditions were included in the manuscript. pH of the media was not measured. “Using the coaxial cable, the DropSens machine was configured for two electrode measurements with one electrode used as the working electrode and the other electrode used as the reference/counter electrode. The cable is an open ended coaxial adaptor with inner and outer conductor electrode dimensions of 2 mm and 5 mm respectively and a length of 7 mm which allows for a sample volume of 500 µl. Both electrodes are made from Nickel. The coaxial cable was secured to ensure stability during measurements. The electrolyte used was the RPMI full media supplemented with 10% FBS”. (Page 7, lines 230-236)
- Figures 4 and 5 could be combined into 1. It’s about surface area coverage of the cells. Are these individual (single) cells or a cluster of cells? From the images in Figure 4, it looks like a cluster of cells (I have no experience here). If it’s a cluster, how is the size of the cluster determined? For example, a cluster of 5 cf with a cluster of 10. How would this affect the results and their interpretation?
Response: The authors feel that combining the images will reduce clarity of particularly Figure 4 as they will have to occupy a reduced area. They are all individual cells. When they differentiate form THP-1 to DCs and macrophages, they change in size and shape and expand while attaching to the surface of the flask.
- Lines 161 and 164: I don’t understand the way the data is being discussed. Why is 0.9V described as maximum positive (and negative) potential? What I see is a reduction peak potential at approximately -0.2V
Response: The justification has been included in the manuscript. “Although reduction peaks at -0.2 V are observed for all experiments, the regions of maximum and minimum potential are of more interest because the peaks correspond to the sample concentrations49.” (Page 10, lines 333-336)
- Line 183: ‘…display a trend’. I don’t follow this either. All the results show a similar pattern. Perhaps the authors would elaborate and rephrase to avoid confusion.
Response: The sentence has been revised to “Comparing the three plots, it was noticed that only THP-1 cells displayed the expected trend of increased capacitance with increasing concentration. Electrochemical sensors react with the analyte under test to produce an electrical signal proportional to the analyte concentration51. The inconsistency with DCs and macrophages was likely due to the lack of a homogenous suspension as cells might not have fully differentiated.” Page 11, lines 359-361)
- Two many figures. Perhaps some of this could be sent to supporting information.
Response: Figure 1 has been removed.
- Line 277: Please clarify ‘accurate data’.
Response: Sentence has been revised to “An optimum Srate of 0.004 V/s was selected which allowed for accurate data, data (this value of Srate limits the Non-Faradic current and therefore background noise which affects the sensitivity of the voltammetry system46) …” (Page 10, lines 209-212)
- Number of replicates and what error bars stand for (SEM or CV or Std) should be included in Figure captions.
Response: Number of replicates and error bar type have been included in all necessary captions. See point 15.
- Figure legends should be stand alone. Please include experimental parameters and conditions in captions.
Response: Figure captions revised to include experimental parameters and conditions where applicable.
Figure 3. Average mean fluorescent intensity of different cell markers for THP-1, DCs, and MACs with S.E.M bars obtained for three measurements. Cultured cells were washed, suspended at 3×10^4 in 200µl cold FACS solution (DPBS; Gibco-Invitrogen) and incubated with FITC- or PE- conjugated monoclonal antibodies or appropriate isotypic controls for 30 minutes. Cells were then washed twice and resuspended in 300 µl of cold FACS solution. Stained cells were analyzed with (BD Accuri C6 plus). Cell debris was excluded from the analysis by setting a gate on forward and side scatter that included only cells that are viable. (Page 8)
Figure 4. (A) THP-1 Immune cells before differentiation. THP-1 was first cultured in RPMI-1640 media then differentiated into DCs and MACs. Human monocytic THP-1 cell line (ATCC, Manassas, VA, USA)35 were cul-tured in RPMI-1640 media supplemented with 10% fetal bovine serum (FBS), 1% sodi-um pyruvate, 0.01% of mercaptoethanol and 1% penicillin/streptomycin at 37 ◦C, 5% CO2 and 95% humidity. (B) DCs and (C) MACs after differentiation respectively. DCs were differentiated based on Berges et al. protocol. To induce differentiation rhIL-4 (200ng=3000 IU/ml) and rhGM-CSF (100 ng/ml =1500 IU/ml), rhTNF-α (20 ng/ml = 2000 IU/ml) and 200 ng/ml ionomycin were added to the FBS- free media. For the macrophanges, the differentiating and activation protocols of THP-1-derived macrophages were adapted and modified from Genin et al.37. THP-1 cells were terminally differentiated into uncommitted macrophages (MPMA) with 300 nM phor-bol 12-myristate 13-acetate (PMA; Sigma-Aldrich, Germany) in RPMI 1640 media without FBS supplement. Afterwards, cells were activated for 48 hours into pro-inflammatory macrophages (MLPS/IFNγ) by adding 10 pg/mL lipopolysac-charide (LPS; Sigma, USA) and 20 ng/mL IFNγ (Biolegend, San Diego, CA, USA), or in-to anti-inflammatory macrophages (MIL-4/IL-13) with 20 ng/mL interleukin 4 (IL-4; Bio-legend, USA) and 20 ng/mL interleukin 13 (IL-13; Biolegend, USA).THP-1 cells have round shape and are suspended in the media, DCs are attached and spread their dendrites in the flask. MACs are also adherent but without the elongations of the DCs. (D), (E) and (F) show the selection done in Image J software for the calculation of the area of the THP-1, DCs and MACs respectively.” (Page 9)
Figure 5. The calculated average area of each cell with S.E.M bars. MACs have the largest area, followed by DCs and lastly THP-1 cells. (Page 10)
Figure 6. I-V curve for the three types of cells using drop sense technology A: THP-1, B: DCs, C: MACs. There are no clear differences between the three graphs. RPMI full media supplemented with 10% FBS used to dilute the cells. It was also used as the media. Measurements were done using a two Nickel electrode configuration, scan range of -0.9 V to 0.9 V and a scan rate of 0.04 V/s. (Page 10)
Figure 7. Current versus time and voltage versus time curves for the three types of cells from the drop sense technology A: THP-1, B: DCs, C: MACs. RPMI full media supplemented with 10% FBS used to dilute the cells. It was also used as the media. Measurements were done using a two Nickel electrode configuration, scan range of -0.9 V to 0.9 V and a scan rate of 0.04 V/s. (Page 11)
Figure 8. Capacitance-time curve for the three types of cells before media de-embedding (removing the value of media from the rest of the samples) A: THP-1, B: DCs, C: MACs. Capacitance values were extracted using MATLAB, basing on the fact that the capacitive current measured is proportional to the rate of change of the applied potential with the constant of proportionality equal to the capacitance. There is no consistent trend between the concentration and capacitance. (Page 11)
Figure 9. Capacitance-time curve for the three types of cells after media de-embedding A: THP-1, B: DCs, C: MACs. De-embedding was performed by diving each of the concentration values in Figure 8 by their corresponding media value. (Page 12)
Figure 10. Capacitance vs concentration for the three types of cells after the de-embedding process. A: Capacitance versus concentration at 29.2 s where maximum capacitance occurs for each of the cell types, B: Average capacitance for the three concentrations with S.EM error bars. MACs have the highest values and DCs have the lowest values consistent with the literature. (Page 12)
- Consistency is terminology. For example, use DC consistently, instead of dendritic or dendritic cells.
Response: All occurrences of dendritic/ dendritic cells after first mention have been replaced with the abbreviation DCs. Also macrophages have been replaced with the MACs abbreviation.
- Methods
- Electrochemical methods have not be well discussed, especially the electrochemical system itself (electrode types, surface areas, electrolyte and similar).
Response: Electrochemical methods details have been added to the methods section (See point 8).
- Please present system optimisation data, at least in SI.
Response: Optimisation data have been included in the supplementary material (Page 13, line 427-444.
- Line 276: The presentation of the results is suboptimal. The results are left hanging, no proper units used. For example, scan rate data should be presented as 0.004V/s. (or 4mV/s). Importance of optimising step potential as well as scan rate should be mentioned.
Response: The units for the scan rate have been updated as requested.
“An optimum Srate of 0.004 V/s was selected which allowed for accurate data, sufficient cur-rent flow and absence of time-dependent charging and discharging effects. This value gave the highest capacitance resolution which can aid with distinguishing between cells.
Secondly both Estep and Srate values were varied simultaneously from 0.009 V to 0.01 V and from 0.009 V/s to 2 V/s respectively. It was found that corresponding low values did not allow for proper current flow and high values of Srate did not allow for sufficient charge of the sample. Additionally equal values of Estep and Srate did not give the correct shape for cyclic voltammetry graph. Hence from the experiments the optimum values of Estep and Srate were selected to be 0.002V and 0.04 V/s respectively.” (Page 6, lines 209-220)
- Line 284: Change ‘voltammeter graph’ to ‘voltammogram’.
Response: Phrase has been replaced as advised. “Additionally equal values of Estep and Srate did not give the correct shape for the cyclic voltammogram” (Page 7, lines 218-219)
- Line 292: ‘… being charged’. Please rephrase. Technically reads poorly.
Response: Sentence rephrased. “It was found that although the screen printed electrode is low cost, disposable and can give results for low volumes, current flow in the samples experiences interference, and as a result, not all cells are charged.” (Page 7, lines 225-227)
Round 2
Reviewer 1 Report
The revised version can be accepted.
Author Response
The respected reviewer did not raise any further issue or concern.
Reviewer 2 Report
The authors' responses were confirmed.
Author Response

(The authors gave the same response as above.)

Reviewer 3 Report
The authors have made effort to address some of the issues raised. Some issues are still outstanding:
1) Peak potential and Current at peak potentials: I believe that the authors are incorrect to note peak potential as 0.9V and -0.9V. There is no peak in sight. The least the authors can do is extend the scan rate and see what happens.
2) What were electrode system for the scree-printed electrodes. What sample size was used?
3) What were the redox species within the sample?
Author Response
Reviewer #3
The authors have made effort to address some of the issues raised. Some issues are still outstanding:
1) Peak potential and Current at peak potentials: I believe that the authors are incorrect to note peak potential as 0.9V and -0.9V. There is no peak in sight. The least the authors can do is extend the scan rate and see what happens.
The scan rate was varied to identify the optimum value taking into account the applied potential range as discussed in the optimization section of the manuscript. Additionally the applied potential range was sufficient to allow for differentiation in behavior between the samples with different concentrations.
The equipment Dropsens is limited in capabilities, the maximum range of applied voltage is from -0.9V and +0.9V. The optimum scan rate has been determined with the minimum value that can provide smooth behavior to compare comparisons among the different type of cells and concentrations. This is one year results and cannot be conducted at the time being, we need to re run the whole design of experiment which will last for another year.
2) What were electrode system for the scree-printed electrodes. What sample size was used?
The screen printed electrodes was used initially to determine the scan rate, the same results obtained with the coaxial adaptor. Strip’s general dimensions: 3.4 x 1.0 x 0.05 cm. Reference electrode and electric contacts made of silver (unless otherwise stated).
3) What were the redox species within the sample?
The manuscript does not report any redox, we preferred not to do the analysis based on the screen printed electrodes and conducted the experiment on bulk electrodes of the coaxial adapter.
We appreciate the respected reviewer # 3 comments, we do apologize for not being able to conduct further experimentation.